# Ecological Response of the Subsidy and Incentive System for Grassland Conservation in China

**Huilong Lin** [1,*], **Yuting Zhao** [2] **and Ghulam Mujtaba Kalhoro** [1]

1   State Key Laboratory of Grassland Agro-Ecosystems, College of Pastoral Agriculture Science and Technology, Lanzhou University, Lanzhou 730020, China; kalhoro2021@lzu.edu.cn
2   State Key Laboratory of Plateau Ecology and Agriculture, College of Agriculture and Animal Husbandry, Qinghai University, Xining 810016, China; zhaoyt14@lzu.edu.cn
*   Correspondence: linhuilong@lzu.edu.cn; Tel.: +86-13993181651

**Abstract:** The overexploitation of Grasslands without any return-back and compensation is the major cause of degradation and deterioration of the grassland ecosystem. The Subsidy and Incentive System for Grassland Conservation (SISGC) in China aimed to restore grassland ecology by the reduction of overgrazing, promoting carrying capacity, and increasing alternative employment of herders in non-husbandry sectors. However, the ecological response to the SISGC still remains unclear on the national scale. Here, we used systematic sampling, and satellite image time series data revealed a widespread proliferation of major ecological indicators for grasslands, contrasting climate and actual net primary productivity (NPP) before (2004–2010) and after (2011–2017) the implementation of SISGC founded the contributions to policy, as simulated by the Carnegie-Ames-Stanford-Approach (CASA) model. On average, by two-phase comparison, the actual grassland NPP increased by 11.72%. The contribution of policy implementation and climate factors increased grassland NPP by up to 61.14% and 38.86%, respectively, but the response of the NPP growth of various grassland types exhibited divergence, mainly divided into policy-led (contribution rate of 52.28–97.02%) and climate-led (contribution rate of 57.56–96.00%). Hence, the SISGC policies should be renewed for different grassland types.

**Keywords:** Subsidy and Incentive System for Grassland Conservation (SISGC); NDVI (normalized difference vegetation index); vegetation coverage; NPP; Comprehensive and Sequential Classification System of grassland (CSCS)

## 1. Introduction

Grasslands are the largest terrestrial ecosystem on earth, accounting for about 40% of the total land area [1]. The net primary productivity (NPP) of grassland is a direct indicator to reflect the health status of vegetation and the ecological quality to a certain extent [2,3]. Grassland is the main terrestrial ecosystem, comprising 41.7% of the land area, of China [4] and is the most basic living resource for native herders, and it provides huge Ecological Services within China and abroad [5,6]. Rapid socio-economic development in China resulted in the degradation and deterioration of grasslands for more than 90% of the total grassland area, which seriously restricted the development of the pastoral economy and even posed a toll on the ecology of the surrounding areas [7].

To reverse this trend, the Chinese central government launched the Subsidy and Incentive System for Grassland Conservation (SISGC) in two consecutive phases (2011–2015 and 2016–2020). Over the past few decades, many countries, such as the United States, the European Union, and China, have undertaken payment-for-ecosystem-services programs to sustain the use of their natural resources [8–10]. Efforts to encourage wild grasslands to return have resulted in a wide range of eco-compensative arrangements, which have been used to fulfill a variety of environmental and ecological goals. Understandings and lessons have been learned from such schemes in the USA, Europe, British, Africa,

Mexican states, and Asia, including China, in extents such as tropical grassy biomes [11], the ecology and conservation of grassland and meadow birds [12,13], future wet grasslands [14], grassland cropping rotations [15], cattle conservation [16], herders' income [17], controlling invasive plant species [18], and the conservation of arthropods [19]. The above studies have emphasized the eco-compensation schemes that have been motivated to discover the way to align different and mutual interests with social benefits towards the management of natural resources. Additionally, they function in a diversity of frameworks and formal sites, where insecurity, distributional concerns, and communal spatiality are of significance. Nevertheless, few countries have created payment-for-ecosystem-services programs for protecting their grasslands on a large scale. From the viewpoints of area scale and total monetary transfers, the SISGC is considered as the world's largest grassland conservation program [20,21]. The first phase involved a cumulative investment of over 10 billion US dollars from the Chinese central government to eight major grassland regions, including Inner Mongolia. Later, another five provinces were added in the second phase, covering 268 pastoral and semi-pastoral counties in the country, with a planned investment of nearly 15 billion US dollars from the central government. The primary objective of the SISGC policy was to restore grassland ecology by the reduction of overgrazing, promoting animal husbandry, and increasing the alternative employment of herders in non-husbandry fields [21,22]. Grassland ecological compensation has a positive impact on household income and the agricultural economy [23–25]. However, the process is still at its infancy. Hu et al. [23] demonstrated that the grassland–livestock balance (one sub-policy of the SISGC) encouraged large farms to reduce sheep stock, which influenced the impact of the implementation of the policy on the decrease of cattle stock present on farms of varying sizes. However, the research on the policy response of grassland ecology to the SISGC is inadequate, especially in China on national scale, because of the vast land area with various types of grasslands [26]. There are substantial external and environmental differences in grasslands of different regions, so there may be significant spatial and temporal variances in the ecological response of different grassland types. How does the ecological response of grasslands in different regions vary with the implementation of the SISGC? If so, what is the changing trend? Are the driving mechanisms the same for each region? None of these questions have been addressed yet, which severely restricts the feedback on the SISGC's implementation to policymakers and affects the policy's further improvement for the future. Since grassland degradation results from both anthropological and natural factors [27], the coupling process of overgrazing, global climate change, rodents, and biological invasion is extremely complex [28,29]. The restoration of grassland degradation can be multifaceted, such as the NDVI (normalized difference vegetation index), coverage, and biomass changes. Because NPP measurement is time-consuming and labor-intensive, it is, hence, difficult to achieve on a large-scale and over a long-term sequence. Keeping in view the environmental conditions of China, to find out a suitable NPP estimation model, using an existing observational data to reveal the spatio-temporal variation characteristics of NPP in response to policy, which can differentiate the respective contribution of human activities and climate factors, will be an important basis to judge the response of grassland ecology to the SISGC.

Research towards the response of grassland ecology to the SISGC is an important basis for decision-making by the central government, implementation by local governments, and the source for rewarding herders. This paper attempts to study the spatial and temporal differentiation of ecology and the driving mechanism of different grassland types in China before and after the implementation of the SISGC to achieve the following objectives: (1) analysis of the spatial and temporal differentiation of the grassland NDVI and grassland coverage in China before (2004–2010) and after the implementation of the SISGC (2011–2017); (2) identify the actual grassland NPP estimation model suitable for China and a comparison based on system sampling data; (3) based on the CSCS (Comprehensive and Sequential Classification System of grassland) [26], analyze the difference between the climate NPP and the actual NPP before and after the implementation of the SISGC and

determine the respective contribution of human activities (mainly the implementation of SISGC) and climate factors to the changes of the grassland NPP.

## 2. Materials and Methods

### 2.1. System Sampling

The system sampling was conducted as a stratified random sampling (SRS) strategy as described by [30] (Figure 1), based on the area of the CSCS actual grassland super-class groups (the top classification unit) [26,31]. Actual grassland was determined by integration of the reclassification of IGBP (International Geosphere-Biosphere Program) data, rendering the IGBP land classification data and NDVI data classification scheme (Table A1, Appendix A) [32], and by superimposing the CSCS database of grassland super-class groups on the IGBP land cover type to obtain a spatial distribution map of the actual grassland (Figure 1).

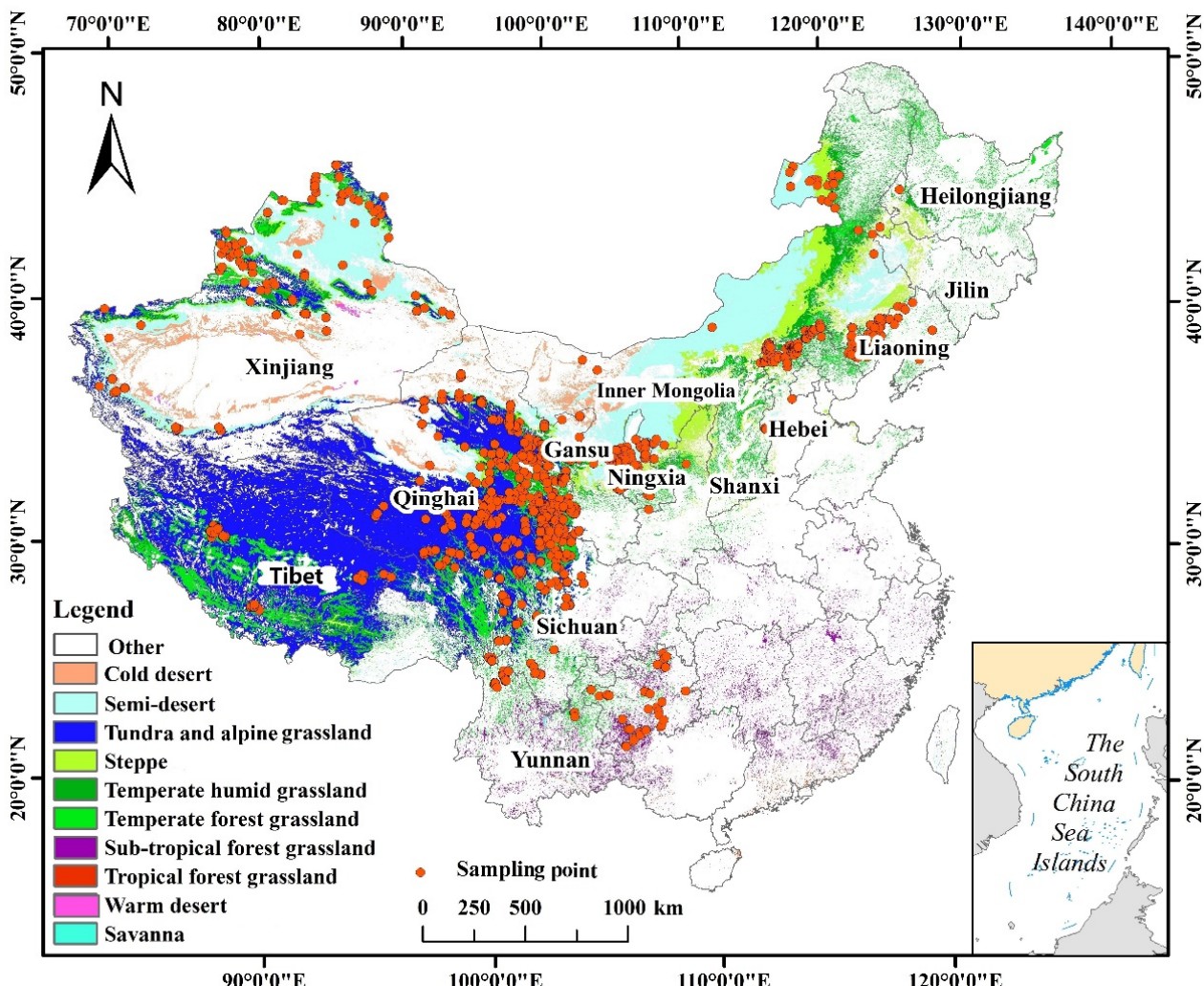

**Figure 1.** Space distribution of the actual grassland CSCS super-class groups in China and system sampling points. Note: The tundra and alpine grassland cover the largest area, accounting for 38.39% of the total grassland area, followed by semi-desert grassland, accounting for 19.91%, and Savanna grassland, which covers the smallest area, accounting for 0.09%.

A total of 2192 sampling points located at seven grassland super-class groups of the CSCS were selected for study (Figure 1). Three sample plots (0.5 × 0.5 m each) were randomly selected for every sampling point during the peak growing season (July–August) for the period of 2004–2015 to obtain the observed NPP values of sampling points [32–37]

(Table A2). However, it was quite difficult to collect samples at 206 sampling points in the Sanjiangyuan region of the Qinghai-Tibet Plateau due to a high altitude, adverse natural conditions, sparse land, inconvenient transportation, mountainous landscape, and wetlands. The conditioned Latin hypercube sampling (cLHS) method [38] was used to set the sampling points to limit the meteorological, topographical, and geomorphological factors; NDVI; soil type; and texture factors to generate cost layers made by the ArcGIS software. The setting of the cLHS sampling points was performed in R Studio 3.5.0 (https://rstudio.com/software (accessed on 1 May 2020)) and the cLHS software package. The NPP unit of the observed value was g·m$^{-2}$ converted to g C m$^{-2}$ by multiplying the conversion factor by 0.45 [37].

*2.2. Data Acquisition*

2.2.1. Meteorological Data

Meteorological data was downloaded from the National Meteorological Science Data Sharing Service Platform (http://data.cma.cn (accessed on 1 May 2020)), including the daily temperature, precipitation, latitude, and longitude of each site from 2004 to 2017. Each site's temperature and precipitation data were consolidated into a monthly average temperature and cumulative precipitation dataset at the temporal resolution. Conversely, China's meteorological data at the temporal resolution of 1000 m were obtained by a Thin-Plate Smoothing Spline (ANUSPLIN 4.36) [39].

2.2.2. MODIS Data

Remote sensing data were gleaned from the consolidated product data of MOD13Q1, MCD12Q1v006, and MOD17A3HGF, developed by NASA'S Earth Observing System Data and Information System, EOSDIS (http://earthdata.nasa.gov (accessed on 1 May 2020)). The MOD13Q1 data product was used to calculate NDVI data from January to December for the period of 2004–2017 [40–42]. The annual NDVI maximum for 2004–2017 was calculated using ArcGIS10.2.2 as the NDVI data for the current year [43]. The MCD12Q1v006 data were consolidated with the NDVI data to generate the range of actual grassland, and the data processing method was the same as that of the MOD13Q1 data products. The MOD17A3HGF provided information about the annual NPP [44,45].

*2.3. Methods*

2.3.1. Analysis of the Characteristics of Temporal and Spatial Change of the NDVI and Vegetation Coverage

The temporal variation in the annual maximum and minimum grassland NDVI from 2004 to 2017 was analyzed with a linear trend model of the slope. When the slope is >0%/yr, the NDVI and coverage will increase, or vice versa [46]. The slope formula is:

$$slope = \frac{n \times \sum_i^n i \times (NDVI \ or \ Cover)_i - \sum_i^n i \sum_1^n (NDVI \ or \ Cover)_i}{n \times \sum_i^n i^2 - (\sum_i^n i)^2} \tag{1}$$

where *n* denotes the total number of years (14); *i* represents the value interval from 2004 to 2017 (1–14); $Cover_i$ and $NDVI_i$ are the maximum grassland coverage and NDVI values at ith year, respectively.

The F-test was conducted in each pixel to analyze the further significance of the temporal variation in the NDVI and coverage. If $F > F_{0.05}$ (1, n − 2), the variation was significant at the 95% confidence level; here, $F_{0.05}$ (1, 12) = 4.75. To overlay the slope and the F-test datasets, the variation in the NDVI and coverage in the entire study area was classified respectively into four categories: significant increase (slope > 0%/yr and F > 4.75), increase (slope > 0%/yr and F < 4.75), decrease (slope < 0%/yr and F < 4.75), and significant decrease (slope < 0%/yr and F > 4.75) [47]. The formula of the F-test is expressed as:

$$F = \frac{R^2(n-1)}{1 - R^2} \tag{2}$$

where $n$ represents the total number of years (14); $R^2$ is the square of the correlation coefficient between the coverage in each pixel and the time series.

It is assumed that the linear, proportion-weighted combination of the NDVI values from bare soil and green vegetation, respectively, contribute to the total NDVI value in a pixel of an image in a pixel dichotomy model. Following this principle, an NDVI pixel observed by remote sensors can be marked as $NDVI_{veg}$ for green vegetation information and $NDVI_{soil}$ for bare soil information [48]. The formula of the pixel dichotomy model is:

$$VC = \frac{(NDVI - NDVI_{soil})}{(NDVI_{veg} - NDVI_{soil})} \tag{3}$$

where $VC$ represents vegetation coverage, $NDVI_{soil}$ is the NDVI value of bare soil, and $NDVI_{veg}$ is the pure pixel NDVI value of a 100% vegetation pixel.

To minimize the unavoidable error of satellite images, the corresponding NDVI values of the 0.5% and 99.5% cumulative percentages were regarded as the $NDVI_{soil}$ (0.5%) and $NDVI_{veg}$ (99.5%) values.

### 2.3.2. Comparison and Selection of the Grassland NPP Models
### CASA (Carnegie-Ames-Stanford-Approach) Model

The CASA model shows a process, based on the principle governing light energy utilization [49], calibrated with more than 1900 measured sites worldwide [50]. The estimated NPP is primarily determined by Absorbed Photosynthesis Active Radiation (APAR) and the conversion rate of solar energy ($\varepsilon$) [51,52]. The formula is as follows:

$$NPP(x, t) = APAR(x, t) \times \varepsilon \tag{4}$$

where $APAR(x, t)$ represents the photosynthetically active radiation (MJ·m$^{-2}$) absorbed by pixel $x$ in month $t$, and $\varepsilon(x, t)$ represents the actual utilization rate of light energy (g C MJ$^{-1}$) of pixel $x$ in month $t$.

$$APAR(x, t) = PAR(x, t) \times FPAR(x, t) \tag{5}$$

where $FPAR(x, t)$ represents the proportion of the vegetation canopy absorbing the photosynthetically active radiation ($PAR$).

When the FPAR parameters were estimated in the CASA model, CSCS was introduced into the model, and the maximum value of NDVI for each grassland type was obtained, whereas the maximum vegetation index ratio for each grassland super-class group was calculated as described by [26,53,54]. Finally, the FPAR for each grassland super-class group was confirmed using a Breathing Earth System Simulator (BESS) as PAR data, stored in an international standard NC format and downloaded from http://environment.snu.ac.kr/bess_rad (accessed on 1 May 2020) [55]. The PAR absorbed by vegetation is affected by temperature and moisture stress [56]. To ensure the reliability and feasibility of the data source, this paper used the ratio between the actual and potential evapotranspiration of the region to reflect the soil moisture [57]. The water stress coefficient was calculated by the actual and potential evapotranspiration, and the temperature stress coefficient was calculated with the monthly average temperature and NDVI data [49,58]. For estimation of the grassland NPP, previous researchers have used the maximum light energy utilization rate of 0.389 g C MJ$^{-1}$ for global vegetation as the monthly maximum light energy utilization rate $\varepsilon_{max}$ [59]. The maximum light energy utilization rate significantly varies among different vegetation types [51]; hence, we used the maximum light energy utilization rate of grassland super-class groups to estimate the NPP [60].

### Model Validation and Comparison

The observed and the estimated NPP values by MOD17A3HGF and CASA model were plotted with the help of a 1:1 line test [61], and the optimal model was determined according

to the model accuracy evaluation index, which included the coefficient of determination ($R^2$), correlation coefficient (r), root mean square error (RMSE), and residual prediction deviation (RPD) [62], of which:

$$RMSE = \sqrt{\frac{\sum_i^n [E(y_i) - y_i]^2}{n}} \tag{6}$$

$$RPD = \frac{SD}{RMSE\sqrt{n/(n-1)}} \tag{7}$$

$E(y_i)$ and $y_i$ are the estimated and observed NPP, respectively, and $n$ is the sample size, whereas *SD* represents the standard deviation of the observed NPP.

2.3.3. Analysis of the Response of the Actual NPP to Human Activity and Climate Factors
Analysis of the Spatial Patterns and Trends of the Climate NPP

Hydrothermal conditions are the most essential factors affecting grassland phenomena and processes [26]. The function of the humidity (K) and the annual accumulated temperature $\geq 0\,°C$ ($\sum\theta$) to express the primary productivity revealed the intrinsic relationship between the grassland type and its NPP. The Classification Indices-based Model (CIM), based on CSCS, has a higher accuracy of potential vegetation simulations at a large scale (national/global) than that of other climate-related models [31,32,34,54,61]. In this study, the CIM was used to calculate the temporal and spatial patterns of climate NPP (2004–2017). The calculation method of the CIM is as follows:

$$NPP = L^2(K)\frac{(0.1 \times \sum\theta \times (K^6 + L(K)K^3 + L^2(K)))}{(K^6 + L^2(K)) \times (K^5 + L(K)K^2)}e^{-\sqrt{13.55 + 3.17K^{-1} - 0.16K^{-2} + 0.0032K^{-3}}}$$

$$L(K) = 0.58802K^3 + 0.50698K^2 - 0.0257081K + 0.0005163874$$

$$K = \frac{R}{0.1 \times \sum\theta} \tag{8}$$

In the formula, $L(K)$ is the equation based on humidity as a parameter, and $R$ is the annual precipitation (mm).

At the same time, an inter-annual dynamic trend analysis of the grassland climate NPP was also conducted, as in Formula (1) [46].

Analysis of the Law and Trend of Spatial and Temporal Zonality of the Actual NPP

The optimal model as previously described in Section 2.3.2. was used for calculating the actual grassland NPP and its spatial and temporal patterns for the period (2004–2017). Meanwhile, the inter-annual dynamic change trend and significance analysis were conducted with the same method as described in Section 2.3.1.

Analysis of the Response of the Actual Grassland NPP to Human Activity and Climate Factors

Climate and human activities are two key factors affecting the grassland NPP. The actual grassland NPP is the result of human activities on the climate NPP [52,63]. Since 1985, China has promulgated and implemented the "Grassland Law", which was later amended and improved several times in 2002, 2009, and 2012 to protect grassland effectively [22]. Since then, there has been no change in the grassland area for 2004–2017. The most significant human activity on grassland was the SISGC; hence, this paper assumes that based on the actual grassland area in 2010, large-scale human activities in 2004–2017 have resulted from the implementation of this policy, i.e.,

$$\Delta NPP_{management\ or\ policy} = NPP_{actual} - NPP_{climate}$$

To quantify and differentiate the driving effect of the SISGC policy implementation and climate factors on the actual NPP changes, it is assumed that the actual volume of change is the sum of the NPP change under the effect of the climate and the NPP triggered by the SISGC policy [64,65], i.e.,

$$\Delta NPP_{actual} = \Delta NPP_{climate} + \Delta\Delta NPP_{management\ or\ policy} \quad (9)$$

Here, $\Delta NPP_{actual}$ and $\Delta NPP_{climate}$ are the changes of the actual and the climate NPP before and after the implementation of the SISGC, respectively. $\Delta\Delta NPP_{management\ or\ policy}$ refers to the change of $\Delta NPP_{management\ or\ policy}$ under the influence of human activities before and after the implementation of the SISGC.

## 3. Results

*3.1. Response of China's Grassland NDVI and Coverage before and after the Implementation of the SISGC*

3.1.1. Spatial Distribution and Change Trend of Grassland NDVI

China's grasslands' NDVI was temporally on the rise before the SISGC, but the annual average NDVI spatially increased by 7.32% after the implementation of the SISGC (Table A3), and the proportion of the grassland NDVI (greater than 0.3) increased from 55.07% (Figure 2a) to 58% (Figure 2b), while the proportion of the NDVI (less than 0.3) was decreased from 44.93% (Figure 2a) to 41.43% (Figure 2b). After the implementation of the SISGC, the NDVI of 72.5% of the grassland area was on the rise, while that of 27.5% of the grassland area was on the decline, showing a slight increasing trend in general (Figure 2c,d).

The response of different grassland super-class groups to NDVI growth varied, with a stable NDVI for the tundra and alpine grassland (Table A3), while the NDVI growth for other grassland super-class groups ranged from 1.35% to 12.50% (Table A3). Among them, the proportion of the increase of the semi-desert grassland NDVI was recorded as the highest, at 81.55%. The proportion of the increased area trend for other super-class groups in the order from high to low was Steppe, sub-tropical forest grassland, cold desert, temperate humid grassland, tropical forest grassland, temperate forest grassland, and Savanna (Table A3).

3.1.2. Spatial Distribution and Change Trend of Grassland Coverage in China

The annual average grassland coverage increased by 5.31% by a two-phase comparison, which increased from 41.39% (before the implementation of the SISGC (2004–2010), Figure 3a) to 43.59% (after the implementation of the SISGC (2011–2017), Figure 3b). Before the implementation of the SISGC, the area with grassland coverage greater than 30% accounted for 55.07% of the total grassland area in China, and the value increased to 58% after the implementation of the SISGC. On the contrary, the area with low grassland coverage decreased (Table A4). After the implementation of the SISGC, 5.4% of grassland coverage decreased significantly ($p < 0.05$), whereas 35.1% of grassland coverage showed a slight decreasing trend ($p < 0.05$), while 47.2% of grassland coverage slightly increased ($p < 0.05$), and 12.3% of grassland coverage significantly increased ($p < 0.05$) (Figure 3c,d).

The coverage responses among grassland super-class groups varied greatly before and after the implementation of the SISGC, ranging from 0.66% to 5.07%. Among them, the maximum increase was 5.07% in the Steppe, and the minimum increase was 0.66% in the tundra and alpine grassland (Table A4).

*3.2. The Optimal NPP Model and Its Application*

3.2.1. Evaluation of Model Accuracy

The CASA model simulation results were more satisfactory than the MOD17A3HGF products: $R^2$ was 0.65 ($p < 0.001$) and 0.26 ($p < 0.001$), respectively; the correlation coefficients were 0.80 and 0.51; the RMSE values were 198.46 and 211.46; and the RPD values were 1.21 and 1.01, respectively (Figure 4, Table A5). It demonstrated that the CASA model had a high

retrieval accuracy for the grassland NPP (Figure 4). As far as the CASA model is concerned, there were differences in the simulation accuracy for different grassland super-class groups, of which the $R^2$ in the simulation and observation value of semi-desert was the highest, ranging from 0.79; the minimum $R^2$ in the simulation and observation value of Steppe was 0.43, and the $R^2$ in the simulation and observation value of other grassland super-class groups ranged between 0.50 and 0.70 (Figure 4, Table A5).

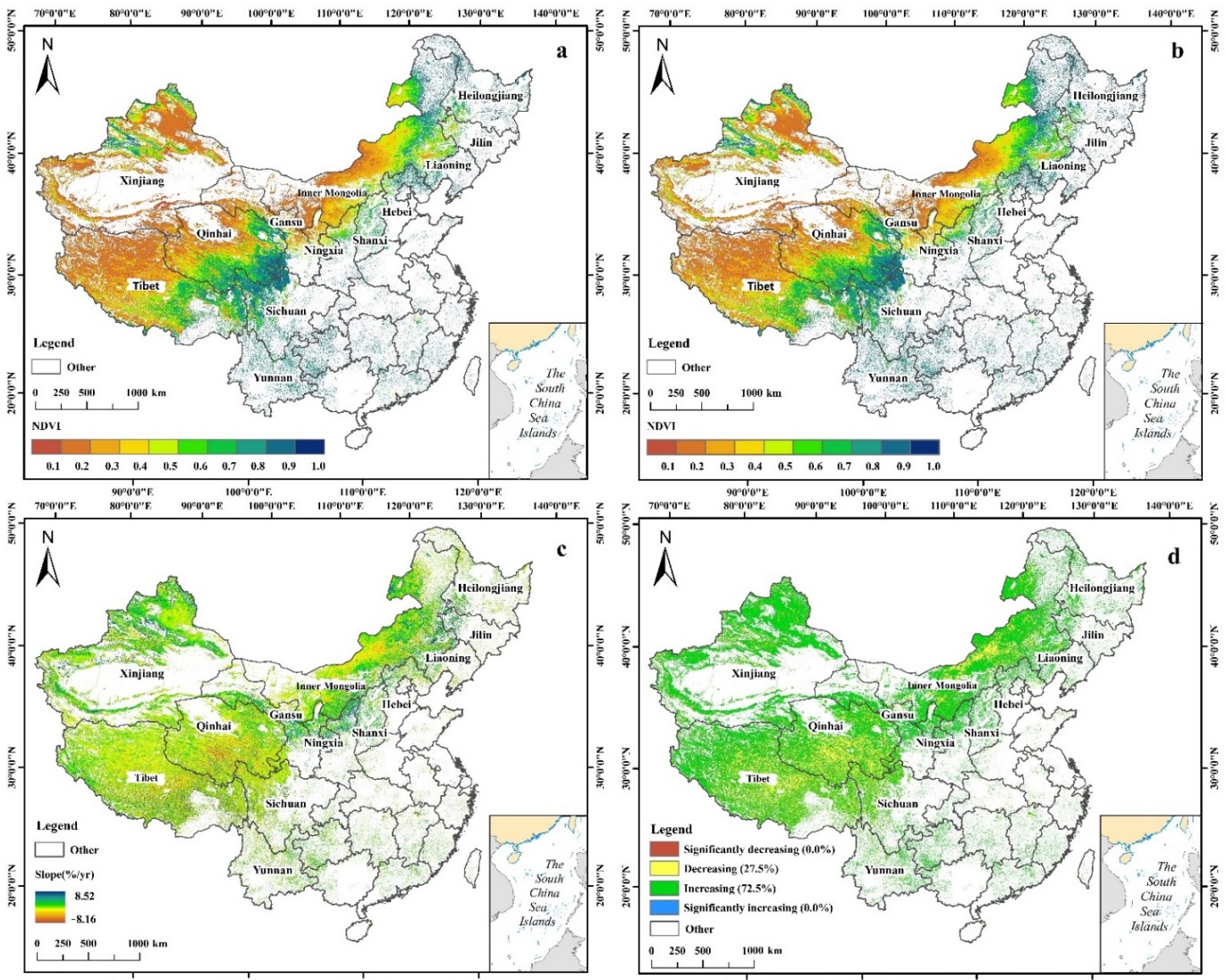

**Figure 2.** The spatial distribution pattern and the changing trend of the grassland NDVI in China before (2004–2010) and after (2011–2017) the implementation of the SISGC. The average grassland NDVI before the SISGC implementation (**a**); the average after the SISGC implementation (**b**); trend changes between 2004–2017 (**c**); trend change grading (**d**).

3.2.2. The Characteristics of the Spatial Change of Actual Grassland NPP before and after the Implementation of SISGC

After the implementation of the SISGC, the high-value area of the actual NPP total value, the annual average, and the change trend increased significantly. The total actual grassland NPP was 497.52 TC g (1 Tg C = $10^{12}$ g C) before the implementation of the SISGC (2004–2010) and 555.83 TC g after the implementation of the SISGC (2011–2017). On average by two-phase comparison, the actual grassland NPP increased by 11.72% from 138.71 g C m$^{-2}$ a$^{-1}$ to 154.96 g C m$^{-2}$ a$^{-1}$ (Figure 5a,b). The significance test of the changing trend of the F-test and slope change trend showed a significant increase in 18.72%,

with a slight increase in 49.84%, a slight decrease in 29.25%, and a significant decrease in 2.19% (Figure 5c,d). The area with a grassland NPP less than 50 g C m$^{-2}$ a$^{-1}$ decreased from 28.41% to 23.78%, while those with a grassland NPP greater than 50 g C m$^{-2}$ a$^{-1}$ increased from 71.59% to 76.22% (Figure 5a,b).

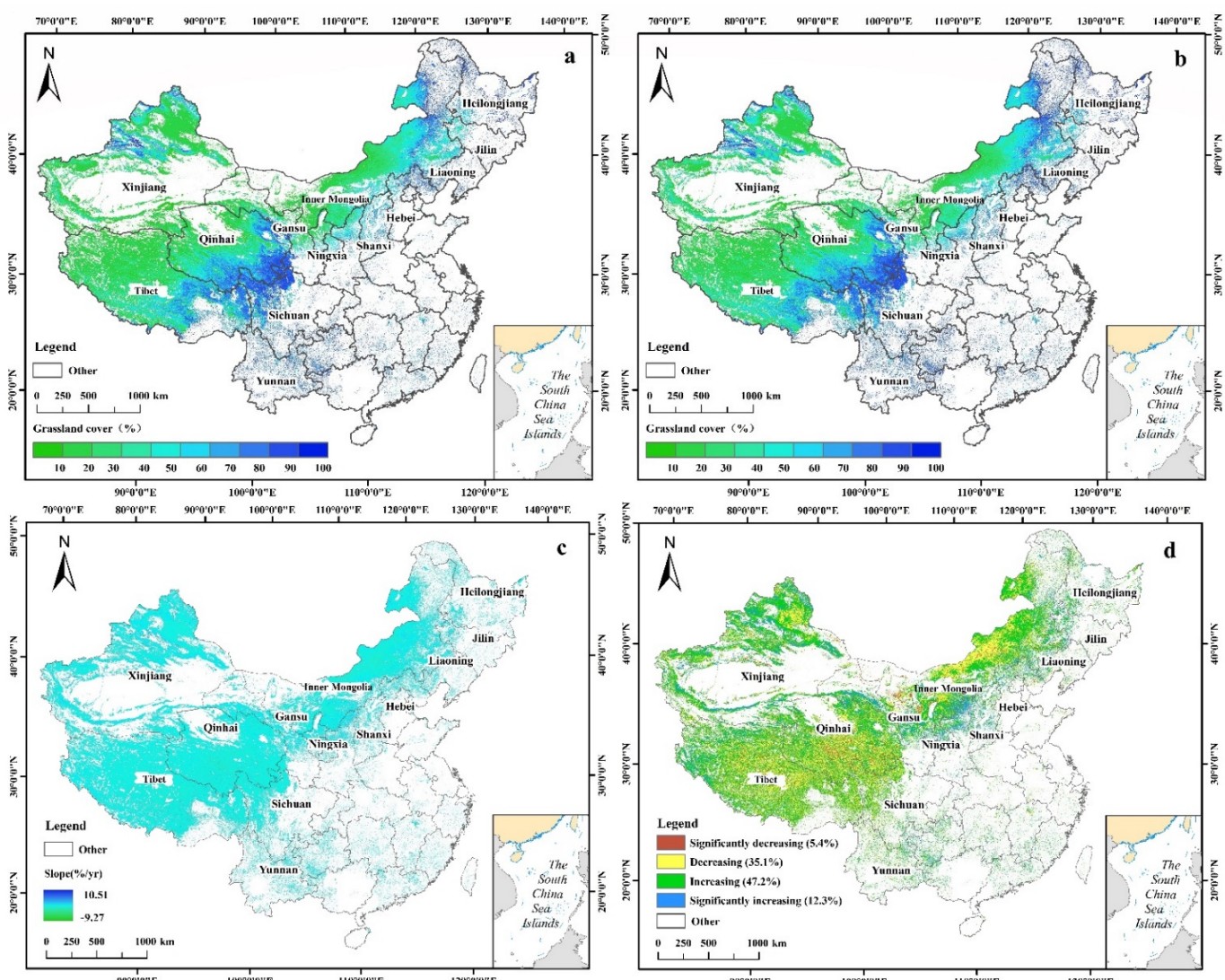

**Figure 3.** The spatial distribution and trends of grassland coverage in China before (2004–2010) and after (2011–2017) the implementation of the SISGC. The average coverage before the SISGC implementation (**a**); the average coverage after the SISGC implementation (**b**); (**c**) trend change (2004–2017); trend change grading (**d**).

The actual NPP among grassland super-class groups before and after the implementation of the SISGC was different. Except for the slight decrease in Savanna (−0.35%), after the policy was implemented, the grassland NPP of all other grassland super-class groups increased by 6.29–23.28% (Table 1). The spatial change trend of the NPP among different grassland super-class groups was obvious after implementing the SISGC, with the warm desert and Savanna grassland NPP presenting a decreasing trend in space (69.53%, 57.02%). The actual grassland NPP of all the other grassland super-class groups showed a 60% increasing trend of space in the area (Table 1).

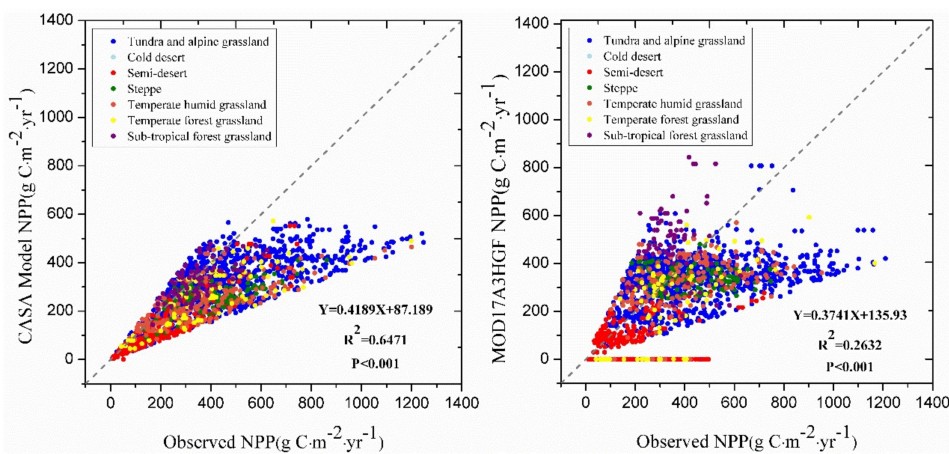

**Figure 4.** The comparison between the observed and estimated NPP values by CASA (**left**) and MOD17A3HGF (**right**).

**Figure 5.** The spatial distribution and trend of the actual grassland NPP in China before (2004–2010) and after (2011–2017) the implementation of the SISGC. The average before the SISGC implementation (**a**); the average after the SISGC implementation (**b**); (**c**) trend change (2004–2017); trend change grading (**d**).

**Table 1.** The statistical description for the average NPP (g C m$^{-2}$ a$^{-1}$) among grassland super-class groups before (2004–2010) and after (2011–2017) the implementation of the SISGC.

| Grassland Super-Class Group | NPP | | | Percentage of the Grassland Area (%) | | | |
|---|---|---|---|---|---|---|---|
| | Before | After | Significant Increase | Increase | Decrease | Significant Decrease |
| Tundra and alpine grassland | 113.52 | 120.66 | 16.13 | 44.05 | 36.88 | 2.95 |
| Cold desert | 33.33 | 39.71 | 18.68 | 20.97 | 52.96 | 7.40 |
| Semi-desert | 61.84 | 74.71 | 17.24 | 54.09 | 27.81 | 0.86 |
| Steppe | 124.26 | 153.19 | 26.07 | 61.27 | 11.91 | 0.75 |
| Temperate humid grassland | 161.47 | 191.29 | 50.07 | 15.10 | 32.39 | 2.43 |
| Temperate forest grassland | 258.20 | 285.30 | 18.39 | 55.15 | 24.76 | 1.70 |
| Sub-tropical forest grassland | 324.60 | 358.12 | 28.01 | 53.34 | 16.57 | 2.08 |
| Tropical forest grassland | 297.65 | 330.75 | 33.47 | 45.86 | 17.27 | 3.40 |
| Warm desert | 31.76 | 36.65 | 16.05 | 14.42 | 59.10 | 10.43 |
| Savanna | 302.34 | 301.26 | 6.43 | 36.55 | 50.24 | 6.78 |

*3.3. Grassland Ecological Responses to SISGC*

3.3.1. The Characteristics of the Spatial Change of the Grassland Climate NPP before and after the Implementation of the SISGC

The climate NPP showed the same trend as the actual NPP, but the increase was less than that of the actual NPP. The total climate NPP was 730.24 TC g before the SISGC (2004–2010) and 769.06 Tg C after the implementation of the policy (2011–2017); meanwhile, the average NPP increased from 203.81 g C m$^{-2}$ a$^{-1}$ to 214.65 g C m$^{-2}$ a$^{-1}$ by 5.32% (Figure 6a,b). The significance of the F-test and slope change trend showed a slight increase of 70.23% and a slight decrease of 29.77% (Figure 6c,d). The area with a grassland climate NPP less than 100 C/m$^2$ a$^{-1}$ decreased from 16.05% to 12.73%, while the area with a grassland NPP greater than 100 C/m$^2$ a$^{-1}$ increased from 83.95% to 87.27% (Figure 6a,b).

The average NPP of almost all grassland super-class groups and the trend of spatial growth were on the rise, with increases ranging from 2.53% to 18.8% and above 70%, respectively (Table 2). Only the average climate NPP of Savanna decreased slightly after the implementation of the SISGC (4.53%), and the grassland climate NPP of Warm desert and Savanna showed a decreasing trend in space by 72.11% and 58.12%, respectively (Table 2). It was obvious that 51.93% of the climate NPP of the tundra and alpine grassland showed a slight increase in space, and 48.07% of the area slightly decreased (Table 2).

**Table 2.** The average climate NPP (g C m$^{-2}$ a$^{-1}$) and trend for all grassland super-class groups before (2004–2010) and after (2011–2017) the implementation of the SISGC.

| Grassland Super-Class Group | NPP | | Percentage of the Grassland Area (%) | |
|---|---|---|---|---|
| | Before | After | Increase | Decrease |
| Tundra and alpine grassland | 177.44 | 181.93 | 51.93 | 48.07 |
| Cold desert | 46.16 | 51.05 | 71.30 | 28.70 |
| Semi-desert | 130.59 | 143.18 | 82.31 | 17.69 |
| Steppe | 196.72 | 222.23 | 94.70 | 5.30 |
| Temperate humid grassland | 191.80 | 209.96 | 85.34 | 14.66 |
| Temperate forest grassland | 270.63 | 281.54 | 74.28 | 25.72 |
| Sub-tropical forest grassland | 658.69 | 684.07 | 89.03 | 10.97 |
| Tropical forest grassland | 866.77 | 891.54 | 93.07 | 6.93 |
| Warm desert | 8.95 | 10.64 | 27.89 | 72.11 |
| Savanna | 427.33 | 407.97 | 41.88 | 58.12 |

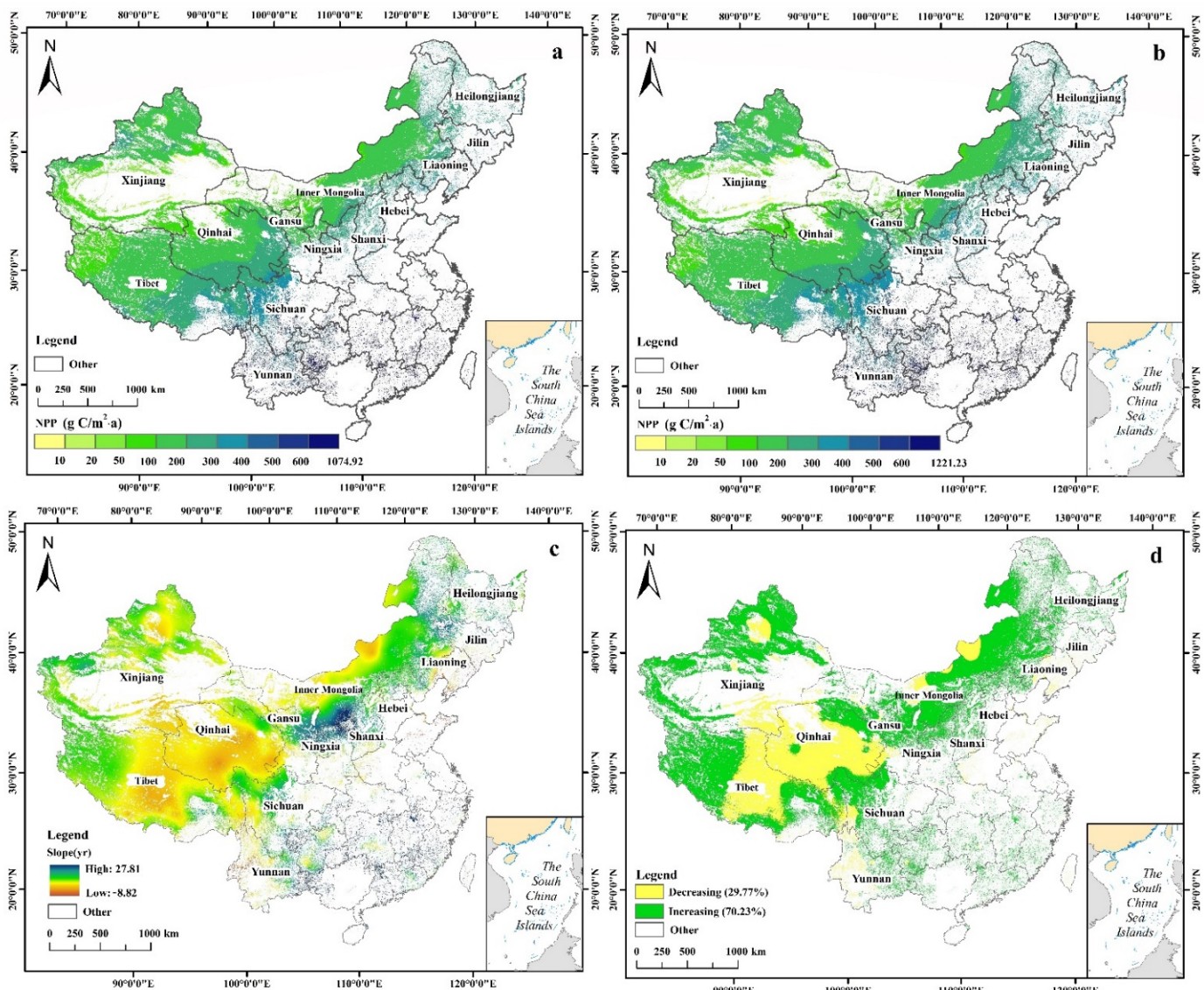

**Figure 6.** The spatial distribution and trend of the grassland climate NPP in China after (2004–2017) and before the implementation of the SISGC policy (2004–2010). (**a**) The average climate NPP (2004–2010); (**b**) the average climate NPP (2011–2017); (**c**) trend change of the climate NPP; (**d**) trend change grading.

3.3.2. The Spatial Characteristics of $\Delta NPP_{management\ or\ policy}$ before and after the Implementation of the SISGC

The negative impact of human activities on the grassland NPP was diminished after the implementation of the policy, and the positive influence was augmented. The total $\Delta NPP_{management\ or\ policy}$ was observed to be −232.71 Tg C before and −213.23 Tg C after policy implementation. The average $\Delta NPP_{management\ or\ policy}$ was −64.95 g C m$^{-2}$ a$^{-1}$ before and –59.52 g C m$^{-2}$ a$^{-1}$ after the implementation of the policy, with an increase of 8.36% in its positive influence (Figure 7). The area with a $\Delta NPP_{management\ or\ policy}$ less than 0 was reduced to 79.71% from 82.41% before the implementation of the policy, while the area with a $\Delta NPP_{management\ or\ policy}$ greater than 0 was increased from 17.59% to 20.29% (Figure 7).

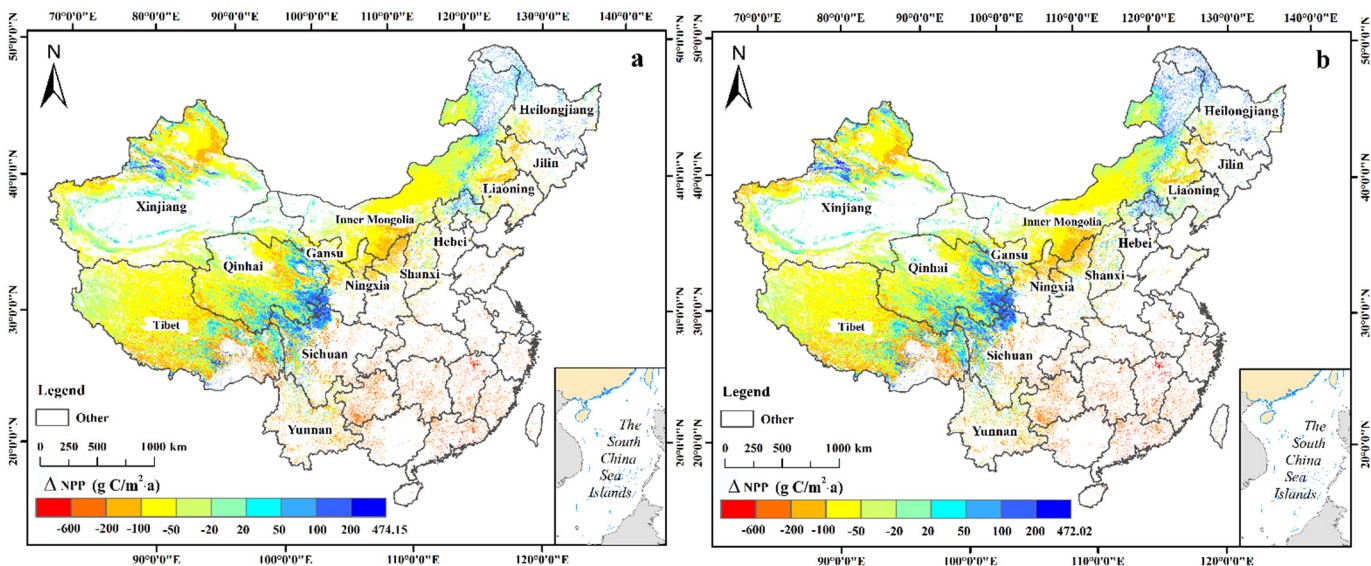

**Figure 7.** The spatial characteristics of the NPP changes caused by human activities before (2004–2010) (left: (**a**)) and after (2011–2017) (right: (**b**)) the implementation of the SISGC policy.

After the SISGC, the positive influence of human activities on all grassland super-class groups was enhanced, with an increase ranging from 0.28 g C m$^{-2}$ a$^{-1}$ to 18.29 g C m$^{-2}$ a$^{-1}$ (Table 3). Among them, the increased area of positive influence in the Temperate humid grassland was the most prevalent, increasing by 6.71%. The increased area of negative influence in the Tropical forest grassland was the lowest, recorded as 0.30% (Table 3).

**Table 3.** The NPP changes (g C m$^{-2}$ a$^{-1}$) caused by human activities before (2004–2010) and after (2011–2017) the implementation of the SISGC.

| Grassland Super-Class Group | $\Delta NPP_{management\ or\ policy}$ | | Percentage of the Grassland Area (%) | | | |
| | | | Before | | After | |
| | Before | After | Negative Effects | Positive Effects | Negative Effects | Positive Effects |
|---|---|---|---|---|---|---|
| Tundra and alpine grassland | −63.65 | −60.98 | 86.33 | 13.67 | 84.76 | 15.24 |
| Cold desert | −12.81 | −11.32 | 69.18 | 30.82 | 67.82 | 32.18 |
| Semi-desert | −68.74 | −68.46 | 97.54 | 2.46 | 95.86 | 4.14 |
| Steppe | −72.44 | −69.01 | 93.71 | 6.29 | 90.70 | 9.30 |
| Temperate humid grassland | −30.31 | −18.65 | 72.29 | 27.71 | 65.59 | 34.41 |
| Temperate forest grassland | −12.35 | 3.85 | 54.55 | 45.45 | 49.07 | 50.93 |
| Sub-tropical forest grassland | −333.98 | −325.83 | 99.01 | 0.99 | 98.30 | 1.70 |
| Tropical forest grassland | −568.69 | −560.42 | 100.00 | 0.00 | 99.70 | 0.30 |
| Warm desert | 22.82 | 26.01 | 19.69 | 80.31 | 18.10 | 81.90 |
| Savanna | −124.91 | −106.62 | 71.95 | 28.05 | 70.69 | 29.31 |

Note: $\Delta NPP_{management\ or\ policy} > 0$ represents positive influence while $NPP_{management\ or\ policy} < 0$ represents a negative influence.

3.3.3. The Respective Contributions of Human Activities (Policies) and Climate Factors to the Actual Grassland NPP Increase

Following policy implementation, the growth rate of the total actual grassland NPP was 11.72%, with human activities contributing 61.14% and climate contributing 38.86%. It has been established that the implementation of the SISGC strategy was the primary contributor in increasing the actual grassland NPP (Figure 8). The increase in the total actual grassland NPP was 58.30 Tg C before and after policy implementation; the increase and growth rate of the total climate NPP was 38.82 Tg C and 5.32%, respectively; and the

total increase and growth rate of the $\Delta NPP_{management\ or\ policy}$ was 19.48 Tg C and 8.37%, respectively.

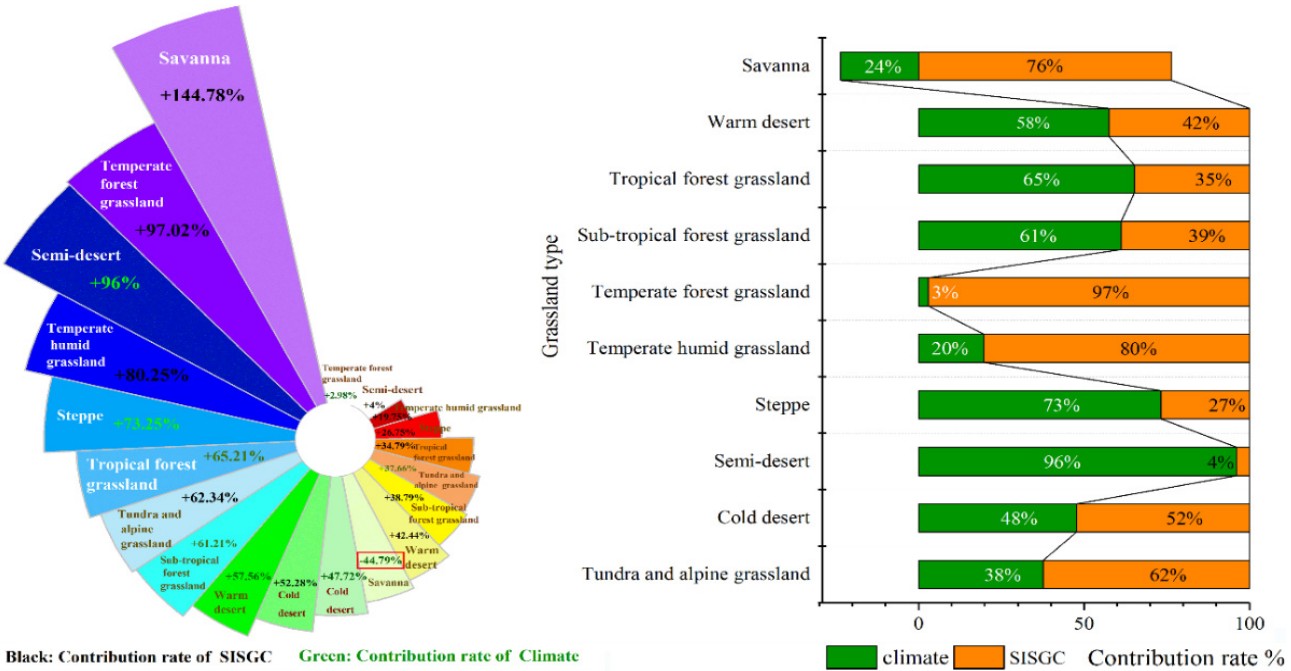

**Figure 8.** The contribution rate of the policy to the increase of the actual NPP among grassland super-class groups after the implementation of the SISGC.

The response of the actual NPP growth for each grassland super-class group was different, mainly divided into policy-led and climate-led. The contribution rate of policy-led human activities ranged from 52.28% to 97.02% (Figure 8), including the Tundra and alpine grasslands, Cold desert, the Temperate humid grassland, and the Temperate forest grassland (Table 4).

**Table 4.** The $\Delta NPP_{actual}$, $\Delta NPP_{climate}$, and $\Delta\Delta NPP_{management\ or\ policy}$ for each grassland super-class group after the implementation of the SISGC.

| Grassland Super-Class Group | Percentage Increase of NPP (%) | | |
|---|---|---|---|
| | $\Delta NPP_{climate}$ | $\Delta\Delta NPP_{management\ or\ policy}$ | $\Delta NPP_{actual}$ |
| Tundra and alpine grassland | 2.53 | 4.19 | 6.73 |
| Cold desert | 10.61 | 11.62 | 22.23 |
| Semi-desert | 9.64 | 0.40 | 10.04 |
| Steppe | 12.97 | 4.74 | 17.70 |
| Temperate humid grassland | 9.47 | 38.46 | 47.93 |
| Temperate forest grassland | 4.03 | 131.16 | 135.19 |
| Sub-tropical forest grassland | 3.85 | 2.44 | 6.29 |
| Tropical forest grassland | 2.86 | 1.52 | 4.38 |
| Warm desert | 18.94 | 13.97 | 32.91 |
| Savanna | −4.53 | 14.65 | 10.11 |

In particular, the climate factors had a negative impact on the actual NPP of Savanna, and the contribution rate of the climate NPP to the actual NPP was −44.79%, while the contribution rate of human activities to the actual NPP of Savanna was 144.79%. The positive influence of human activities was the main driving force leading to the increase of the actual NPP of Savanna by 10.11%. The climate factors were the main driving force for the increase in the climate-led actual grassland NPP, with the contribution rate of climate factors ranging from 57.56–96.00%, including five grassland super-class groups, i.e.,

the Semi-desert, Steppe, Sub-tropical forest grassland, the Tropical forest grassland, and the Warm desert grassland (Figure 8).

## 4. Discussion

*4.1. The Improvement of the CASA Model's Parameters May Enhance the Estimation Accuracy of China's Grassland NPP*

The CSCS-based parameter improvements were more consistent with the actual situation of the grassland in China. Zheng [49] used the CASA model to calculate the NPP of various vegetation types in the Qinghai-Tibet Plateau. The ratio of the aboveground and belowground biomass was at a constant value of 3.18, rather than the root–top ratio among different vegetation types, which was greater than 3.18 for all grassland super-class groups (Table A2). According to the CASA verification results, the grassland NPP proposed by Zheng et al. should be less than the simulation value of this paper (141.735 g C m$^{-2}$ a$^{-1}$). However, their simulated average NPP value reached 232.25 g C m$^{-2}$ a$^{-1}$. That may be the reason for this difference, as Zheng et al. calculated the NPP by utilizing the average level of all vegetation types in the Qinghai-Tibet Plateau. Ye et al. [66] estimated the annual NPP in the Poyang Lake floodplain wetland as 348.06 g C m$^{-2}$ a$^{-1}$ using the CASA model, which is greater than the annual average NPP of the present study (116.29 g C m$^{-2}$ a$^{-1}$). This was because the maximum light energy utilization value selected by Ye et al. was 1.054 g C MJ$^{-1}$, greater than that of our study of 0.542 g C MJ$^{-1}$. Moreover, the spatial distribution of the solar radiation was obtained by linear interpolation of the calculated radiation data of each station. The PAR data in this research was calculated by multiplying the solar radiation data obtained by the linear interpolation method by a coefficient of 0.5, without considering the key factor of the solar height angle, which might increase the error of PAR. Xu et al. [67] calculated the NPP for all vegetation types in the Qinghai-Tibet Plateau using the CASA model, with an average NPP of 121.9 g C m$^{-2}$ a$^{-1}$ being close to our average NPP of 141.74 g C m$^{-2}$ a$^{-1}$ in the Qinghai-Tibet Plateau, and they also calculated the FPAR by categorizing all vegetation types of the Qinghai-Tibet Plateau as one type but did not classify them according to different vegetation types, which might have caused a slight deviation from the results of this paper. The total NPP of China's natural grasslands calculated by Zhang et al. [51] using the CASA model was 4.90 Pg C, which is greater than values of present study (555.83 T g C). Zhang et al. [51] studied and estimated the potential grassland NPP under various land-use types, and the exact area of the definite grasslands was not taken as the research object, resulting in a relatively high estimated value. Yang et al. [68] estimated the average NPP of grasslands in Xinjiang to be 129.19 g C m$^2$ a$^{-1}$ by CASA, which was greater than our simulated result of 77.84 g C m$^2$ a$^{-1}$ in the Xinjiang Autonomous Region. The reason for this deviation might be because Yang et al. classified all grassland types into a general category, which caused some errors in calculating the FPAR. The average grassland NPP in Gansu Province, estimated by Liu et al. [69] based on the CASA model, was 380.12 g C m$^{-2}$ a$^{-1}$, which is significantly higher than 150.10 g C m$^2$ a$^{-1}$ of the present study in Gansu Province. This was because Liu et al. [69] only studied the tundra and alpine grasslands and the semi-desert grasslands in Northwestern Gansu Province, excluding the semi-desert and cold desert in the East and North from the grassland range, so the NPP was relatively high. In summary, combined with the CSCS grassland super-class group, the estimated NPP based on the actual grassland in China was more consistent with the actual situation.

The combination of grassland type and land use was more accurate for the real time situation. For example, 235 of 2192 observed sample points in this paper belonged to the Semi-desert and Cold desert grassland super-class group under CSCS. The observed NPP values of these 235 samples ranged from 10.64 to 496.31 g C m$^{-2}$ a$^{-1}$, and the estimated NPP by the CSCS-based CASA model ranged from 0 to 481.43 g C m$^{-2}$ a$^{-1}$. However, the MOD17A3HGF products observed them directly using the International Geosphere-Biosphere Programme (IGBP) land-use type as "unclassified", "permanent wetlands", and "tundra", mechanically assigning a value of 0 to them (Figure 4). Moreover, the

NPP calculated by the MOD17A3HGF based on the IGBP land-use type exaggerated the average value of the grassland NPP and reduced the total NPP because of the outliers. The combination of the IGBP and the CSCS (Table A1) has obvious advantages in the identification of actual grassland, which made up for the deficiency of the MOD17A3HGF products to more accurately respond to the reality of the grassland NPP in China.

*4.2. China's Grassland Ecological Improvement Is Consistent with the Policy Expectations of the SISGC*

The SISGC is designed to balance livestock production, herders' livelihoods, and grassland ecological protection with the core objective of restoring grassland ecology. Hu et al. [23] found that herders' livestock production structures and grazing decisions in the Inner Mongolia Autonomous Region were not affected by the SISGC. In contrast, the livestock market price was the primary factor. It has been argued that livestock production and grazing intensity have not changed due to the implementation of the SISGC policy, and the grassland ecology has not been improved yet. However, Liu et al. [22] indicated that implementing the SISGC policy has reduced the number of sheep and livestock in Inner Mongolia, which could prevent grassland degradation. Byrne et al. [70] noted that the implementation of the SISGC policy in the Ulanqab prefecture in Inner Mongolia did not reduce the stocking rate, that livestock numbers remained at the overgrazing level, and that current policy subsidies were insufficient to offset the economic losses caused by reduced grazing intensity. This paper mainly addressed the issue of whether grassland ecology has been restored after the implementation of the SISGC policy. Three ecological indicators, i.e., NDVI (↑7.32%), coverage (↑5.31%), NPP (↑11.72%), exhibited an upward trend after the implementation of the SISGC policy. Still, the response level of various grassland super-class groups was different. The implementation of the SISGC policy has promoted the NPP in varying degrees (↑4~97.02%) between the grassland super-class groups. For the Tundra and alpine grasslands, the implementation of the SISGC contributed 62.34% to the increase of 6.73% of the actual grassland NPP, owing to the strict implementation of the SISGC and the local government's ecological consciousness regarding the Qinghai-Tibet Plateau as an important ecological shelter in the world's third pole [71]. This was even though the Semi-desert and Steppe's grassland were within the scope of the SISGC implementation. Still, the driving forces of the increased NPP were mainly climatic factors, with the SISGC policy playing a complementary role.

*4.3. The SISGC Should Be Renewed in a Differentiated Manner Based on the CSCS Grassland Super-Class Group*

The compensation standards of the SISGC are confusing and conflicting among the grassland super-class group. The government has formulated an SISGC policy in each province within the scope of the policy. At present, the policy has been implemented depending on the size of the grassland area, excluding ecological factors such as the grassland super-class group and ecological function in most provinces and regions. The policy standards in the Tundra and the alpine grassland located in the Qinghai-Tibet Plateau area are different as they belong to different administrative divisions. The subsidy amount for the grazing ban (sub-policy of the SISGC) in the Qinghai-Tibet Plateau of Gansu Province is 48.95 US dollars ha$^{-1}$. The subsidy amount for the grazing ban on grassland in the Qinghai-Tibet Plateau of Qinghai Province varies, with 8.13 US dollars ha$^{-1}$ in Haixi zhou, 39.53 US dollars ha$^{-1}$ in Huangnan zhou, 14.46 US dollars ha$^{-1}$ in Golo and Yushu zhou, and 28.01 US dollars ha$^{-1}$ in Hainan and Haibei zhou; the national standard of subsidy for the grazing ban has been implemented across the Tibet Autonomous Region at 16.94 US dollars ha$^{-1}$; the subsidy standard in different administrative regions results in a three-fold difference between the minimum and the maximum amount of subsidy for the grazing ban in the Tundra and alpine grassland. As for the desert grassland, the subsidy amount for the grazing ban in Gansu Province is 8.74 US dollars ha$^{-1}$, and in Xinjiang, it is 13.55 US dollars ha$^{-1}$, which is a difference of nearly 1.6 times. The same standard of the subsidy amount for the grazing ban applies in the alpine grassland, desert

grassland, and other grassland types in eight provinces of Ningxia, Sichuan, Yunnan Shanxi, Heilongjiang, Jilin, Liaoning, and Hebei (16.94 US dollars ha$^{-1}$) (Table A6). Different policy implementation programs have resulted in apparent differentiated policy effects. Qinghai Province and the Tibet Autonomous Region are located in the Qinghai-Tibet Plateau region. The grassland super-class groups are mainly in the Tundra and alpine grasslands. Still, a comprehensive SISGC policy has been implemented in Qinghai Province with the consideration of ecological services. In contrast, a single SISGC policy without differentiation has been implemented in Tibet (national standard: grassland and livestock balance of 5.65 US dollars ha$^{-1}$, ban on grazing: 16.94 US dollars ha$^{-1}$). Our results have shown that the contribution rate of human activities to the grassland NPP increase in Qinghai Province is greater than that in Tibet.

A differentiated manner based on the CSCS grassland super-class group would be an important direction for future policy improvement. Grasslands are the basic productive resources for human existence and development and provide diversified ecological services, such as 83% of carbon sequestration and 86% of climate regulation in China [72]. There are significant differences in the ecological services of air protection, soil decontamination, and conservation, etc. [72,73]. In the process of implementing the SISGC, we should not only consider the economic factors, such as the production and livestock-carrying capacity of grassland, but also focus on its ecological services among grassland super-class groups. The compensation standard should be adjusted based on grassland super-class groups and their ecological services. At the same time, wages and the number of grassland ecological supervisors should be increased, the cost of herders should be raised, and the effect of the SISGC regulation by using drones and wireless monitoring equipment should be strengthened. In this way, the grassland ecological security barrier can be built, and the situation can be rehabilitated in the pastoral areas where humans and nature exist in harmony.

## 5. Conclusions

The verification results of 2192 observed NPP data obtained by system sampling have shown that this paper has effectively and reliably improved the CASA model.

Ecological indicators of various grassland super-class groups responded differently to the policy's implementation, with the annual average of the grassland NDVI and coverage in China increasing by 7.32% and 5.31%, respectively. The increase of the NDVI value for each grassland super-class group ranged between 1.35% and 12.50%, and the increase in coverage for each super-class group was between 0.66% and 5.07%. The increase of the average NPP after the implementation of the policy was 11.72%. Except for the slight decrease of the actual NPP of Savanna, the NPP of all other grassland super-class groups increased by 6.29% to 23.28%.

The contribution rate of policy impact and climate factors to the NPP increase was 61.14% and 38.86%, respectively. However, the response of different grassland super-class groups differs, which can be divided into policy-led and climate-led. Previously, the contribution rate of human activities was 52.28–97.02% and 57.56–96.00% after policy implementation.

The results of the present study reveal that the effects of the SISGC proved to be positive in the prevention of grassland deprivation by the provision of subsidies and incentives to pastoral areas. The investigation validates that the program has increased the grassland NPP and the significance of constructing certain programs that are flexible and easily adaptable by indigenous resources' circumstances. The perceptions of the present study could be particularly supportive in advising conservation policy in other developing countries with such degraded grasslands. Consequently, flexible and sustainable practices must be essential for the conservation of grasslands to combat climate change prospects. These concerted struggles are prerequisites to implicate farmers, scientists, and policymakers employed collectively to magnify decision-making to achieve the 21st century goals set for grassland ecology and conservation.

**Author Contributions:** Conceptualization, H.L. and Y.Z.; Writing—review and editing, H.L., Y.Z. and G.M.K. All authors have read and agreed to the published version of the manuscript.

**Funding:** This research received funding from the National Natural Science Foundation of China (32171680 and 31772666).

**Institutional Review Board Statement:** Not applicable.

**Informed Consent Statement:** Not applicable.

**Data Availability Statement:** The data presented in this study are available upon request from the corresponding author.

**Acknowledgments:** The constructive comments and suggestions from reviewers are highly appreciated.

**Conflicts of Interest:** The authors declare no conflict of interest.

## Appendix A

**Table A1.** Reclassified scheme of the MCD12Q1 land cover product.

| Reclassify Type (New Code) | Land Cover Type (IGBP Code) |
|---|---|
| Water bodies (1) | Water bodies (0) |
| Forest (2) | Evergreen needle leaf forest (1), Evergreen broadleaf forest (2), Deciduous needle leaf forest (3), Deciduous broadleaf forest (4), Mixed forest (5), Closed shrub lands (6), Sparse woodland (18) |
| Grasslands (3) | Open shrub lands (7), Woody Savannas (8), Savannas (9), Grasslands (10), Permanent wetlands (11), Desert grassland (16), Sparse savanna (8) |
| Artificial surfaces (4) | Croplands (12), Urban and built-up (13), Cropland mosaics (14) |
| Permanent snow and ice (5) | Snow/Ice (15) |

**Table A2.** The ratio of belowground to aboveground biomass for different grassland classes.

| Grassland Class | Ratio |
|---|---|
| Temperate meadow_steppe | 5.26 |
| Temperate steppe | 4.25 |
| Temperate desert_steppe | 7.89 |
| High cold meadow steppe | 7.91 |
| High_cold steppe | 4.25 |
| High_cold desert steppe | 7.89 |
| Temperate steppe_desert | 7.89 |
| Temperate desert | 7.89 |
| High_cold desert | 7.89 |
| Tropical herbosa | 4.42 |
| Tropical shrub herbosa | 4.42 |
| Warm_temperate herbosa | 4.42 |
| Warm_temperate shrub herbosa | 4.42 |
| Lowland meadow | 6.31 |
| Temperate montane meadows | 6.31 |
| Alpine meadow | 7.92 |
| Marsh | 15.68 |

**Table A3.** The statistical description of the annual average NDVI before (2004–2010) and after (2011–2017) the implementation of the SISGC.

| Grassland Super-Class Group | NDVI | | Percentage of the Grassland Area (%) | |
|---|---|---|---|---|
| | Before | After | Increase | Decrease |
| Tundra and alpine grassland | 0.36 | 0.36 | 62.55 | 37.45 |
| Cold desert | 0.16 | 0.18 | 80.64 | 19.36 |

**Table A3.** *Cont.*

| Grassland Super-Class Group | NDVI | | Percentage of the Grassland Area (%) | |
|---|---|---|---|---|
| | Before | After | Increase | Decrease |
| Semi-desert | 0.25 | 0.28 | 81.55 | 18.45 |
| Steppe | 0.45 | 0.50 | 81.12 | 18.88 |
| Temperate humid grassland | 0.51 | 0.55 | 80.41 | 19.59 |
| Temperate forest grassland | 0.65 | 0.67 | 72.39 | 27.61 |
| Sub-tropical forest grassland | 0.78 | 0.80 | 80.79 | 19.21 |
| Tropical forest grassland | 0.72 | 0.75 | 79.90 | 20.10 |
| Warm desert | 0.17 | 0.19 | 73.37 | 26.63 |
| Savanna | 0.74 | 0.75 | 71.28 | 28.72 |

**Table A4.** The statistical description of grassland coverage in China before (2004–2010) and after (2011–2017) the implementation of the SISGC.

| Grassland Super-Class Group | Annual Average Coverage (%) | | Significant Change | | Slight Change | |
|---|---|---|---|---|---|---|
| | 2004–2010 | 2011–2017 | Increase (%) | Decrease (%) | Increase (%) | Decrease (%) |
| Tundra and alpine grassland | 35.56 | 36.22 | 8.38 | 6.76 | 43.65 | 41.21 |
| Cold desert | 15.55 | 18.22 | 13.80 | 14.49 | 36.82 | 34.90 |
| Semi-desert | 24.79 | 28.20 | 10.73 | 3.68 | 48.65 | 36.94 |
| Steppe | 44.95 | 50.02 | 20.59 | 2.32 | 55.26 | 21.83 |
| Temperate humid grassland | 51.42 | 55.27 | 17.27 | 3.11 | 54.27 | 25.35 |
| Temperate forest grassland | 64.56 | 66.59 | 14.50 | 4.80 | 47.84 | 32.85 |
| Sub-tropical forest grassland | 78.15 | 80.47 | 20.07 | 2.92 | 54.32 | 22.69 |
| Tropical forest grassland | 72.39 | 75.48 | 20.47 | 6.64 | 44.95 | 27.95 |
| Warm desert | 16.62 | 18.85 | 18.67 | 14.68 | 33.77 | 32.87 |
| Savanna | 74.26 | 75.32 | 12.94 | 4.00 | 49.76 | 33.29 |

**Table A5.** The comparison between the observed and estimated NPP values by the CASA model and the MOD17A3HGF product.

| Grassland Super-Class Group | CASA Model | | MOD17A3HGF | |
|---|---|---|---|---|
| | $R^2$ | R | $R^2$ | R |
| Tundra and alpine grassland | 0.51 | 0.71 | 0.14 | 0.37 |
| Cold desert | 0.50 | 0.71 | 0.29 | 0.54 |
| Semi-desert | 0.79 | 0.89 | 0.49 | 0.70 |
| Steppe | 0.44 | 0.66 | 0.11 | 0.33 |
| Temperate humid grassland | 0.52 | 0.72 | 0.21 | 0.46 |
| Temperate forest grassland | 0.50 | 0.71 | 0.39 | 0.62 |
| Sub-tropical forest grassland | 0.47 | 0.69 | 0.16 | 0.40 |

**Table A6.** Two phases of the subsidy and incentive standards of the SISGC for grazing prohibition.

| Province | The Allowance Standard for Grazing Prohibition | |
|---|---|---|
| | 2011–2015 Year | 2016–Now |
| Gansu | Desert grassland: 4.97 US dollars ha$^{-1}$<br>Loess Plateau: 6.66 US dollars ha$^{-1}$<br>Qinghai-Tibet Plateau: 45.17 US dollars ha$^{-1}$ | desert grassland: 8.74 US dollars ha$^{-1}$<br>Loess Plateau: 10.43 US dollars ha$^{-1}$<br>Qinghai-Tibet Plateau: 48.95 US dollars ha$^{-1}$ |

**Table A6.** *Cont.*

| Province | The Allowance Standard for Grazing Prohibition | |
|---|---|---|
| | **2011–2015 Year** | **2016–Now** |
| Xinjiang | Water conservation region: 112.93 US dollars ha$^{-1}$<br>Desert grassland: 12.42 US dollars ha$^{-1}$ | Water conservation region 112.93 US dollars ha$^{-1}$<br>Desert grassland: 13.55 US dollars ha$^{-1}$ |
| Qinghai | Haixi: 6.78 US dollars ha$^{-1}$<br>Huangnan: 31.62 US dollars ha$^{-1}$<br>Golo,Yushu: 11.29 US dollars ha$^{-1}$<br>Hainan,Haibei: 22.59 US dollars ha$^{-1}$ | Haixi: 8.13 US dollars ha$^{-1}$<br>Huangnan: 39.53 US dollars ha$^{-1}$<br>Golo,Yushu: 14.46 US dollars ha$^{-1}$<br>Hainan,Haibei: 28.01 US dollars ha$^{-1}$ |
| Inner Mongolia | 13.55 US dollars standard ha $^{-1}$ | 16.94 US dollars standard ha $^{-1}$ |
| Ningxia | 13.55 US dollars ha$^{-1}$ | 16.94 US dollars ha$^{-1}$ |
| Tibet | 13.55 US dollars ha$^{-1}$ | 16.94 US dollars ha$^{-1}$ |
| Sichuan | 13.55 US dollars ha$^{-1}$ | 16. 94US dollars ha$^{-1}$ |
| Yunnan | 13.55 US dollars ha$^{-1}$ | 16.94 US dollars ha$^{-1}$ |
| Hebei | No implementation of the SISGC | 16.94 US dollars ha$^{-1}$ |
| Shanxi | No implementation of the SISGC | 16.94 US dollars ha$^{-1}$ |
| Liaoning | No implementation of the SISGC | 16.94 US dollars ha$^{-1}$ |
| Jilin | No implementation of the SISGC | 16.94 US dollars ha$^{-1}$ |
| Heilongjiang | No implementation of the SISGC | 16.94 US dollars ha$^{-1}$ |

Note: The allowance standard for grazing prohibition was obtained by organizing official documents on the Subsidy and Incentive System for Grassland Conservation (2011–2015) and the new round of the Subsidy and Incentive System for Grassland Conservation (2016–2020).

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
