# Peer review of "Ecological Response of the Subsidy and Incentive System for Grassland Conservation in China"

_land, doi:10.3390/land11030358_

Round 1
Reviewer 1 Report
Thank you for providing your manuscript to the Land journal. Generally, the manuscript fits into the journal, it respects Scientifc Best Practice and it is written in a good style.
I do not have to many comments, however, the authors must be aware that the manuscript now has a complete China focus which leads to the situation that it is not interesting to the rest of the world scientific audience.
I strongly recommend to improve the introduction with references that refer also to other regions in the world and make a reeference there.
From my point of view, the discussion part is too long and the conclusions part to short and too specific, which must be somehow balanced. In the conclusions, in addition to summarising the actions taken and results, please strengthen the explanation of their significance. It is recommended to use quantitative reasoning comparing with appropriate benchmarks, especially those stemming from previous work, also from outside China.
Author Response
Dear responsible reviewer:
Thank you very much for your useful comments to improve our manuscript. We have modified our manuscript considering your comments, the changes made can be viewed in the track changed file submitted with revision. Below is the point-by-point response in compliance with major comments.
Comment-1: I do not have to many comments, however, the authors must be aware that the manuscript now has a complete China focus which leads to the situation that it is not interesting to the rest of the world scientific audience.
Response: Thanks for the valuable point. In this context, we would like to emphasis that present study is a novel and interesting for readers across the world and can be a reference for other countries to undertake large and high profile payments for-ecosystem services programs to sustain the use of their natural resources. Present research mainly focuses on china because grasslands are the largest terrestrial ecosystem in China, accounting for 41.7% of the land area, and is the most basic living resource for the herders, which provides China and the world with huge ecological and economic resources. To protect and restore the ecosystems, China has implemented several conservation programs and grassland ecosystem subsides and award Schemes.
Comment-2: I strongly recommend to improve the introduction with references that refer also to other regions in the world and make a reference there.
Response: Thanks, we have now incorporated some additional information related to other regions of the world with proper reference in the introduction section (Line 40-57), and the following literature is also supplemented in the revised manuscript:
- Whittingham, M. J. The future of agri‐environment schemes: biodiversity gains and ecosystem service delivery ? Journal of Applied Ecology, 2011,48,509–513.
- Kleijn,D.; Sutherland,W.J. How effective are European agri-environment schemes inconserving and promoting biodiversity? Journal of Applied Ecology,2003,40(6),947–969.
- Deng, L.; Shangguan, Z.; Li, R. Effects of the Grain-for-Green program on soil erosion in China. International Journal of Sediment Research 2012, 27(1) ,120–127. doi:10.1016/S1001-6279(12)60021-3.
- Lehmann, C.E.; Parr, C.L. Tropical grassy biomes: linking ecology, human use and conservation. 2016, 371, 20160329.
- Azpiroz, A.B.; Isacch, J.P.; Dias, R.A.; Di Giacomo, A.S.; Fontana, C.S.; Palarea, C.M. Ecology and conservation of grassland birds in southeastern South America: a review. Journal of Field Ornithology 2012, 83, 217-246.
- Assandri, G.; Bogliani, G.; Pedrini, P.; Brambilla, M. Species-specific responses to habitat and livestock management call for carefully targeted conservation strategies for declining meadow birds. Journal for Nature Conservation 2019, 52, 125757.
- Joyce, C.B.; Simpson, M.; Casanova, M. Future wet grasslands: ecological implications of climate change. Ecosystem Health and Sustainability 2016, 2, e01240.
- Lemaire, G.; Gastal, F.; Franzluebbers, A.; Chabbi, A. Grassland–cropping rotations: an avenue for agricultural diversification to reconcile high production with environmental quality. Environmental management 2015, 56, 1065-1077.
- Sanderson, J.S.; Beutler, C.; Brown, J.R.; Burke, I.; Chapman, T.; Conant, R.T.; Derner, J.D.; Easter, M.; Fuhlendorf, S.D.; Grissom, G. Cattle, conservation, and carbon in the western Great Plains. Journal of Soil and Water Conservation 2020, 75, 5A-12A.
- Gao, L.; Kinnucan, H.W.; Zhang, Y.; Qiao, G. The effects of a subsidy for grassland protection on livestock numbers, grazing intensity, and herders’ income in Inner Mongolia. Land Use Policy 2016, 54, 302-312.
- Weidlich, E.W.; Flórido, F.G.; Sorrini, T.B.; Brancalion, P.H. Controlling invasive plant species in ecological restoration: A global review. Journal of Applied Ecology 2020, 57, 1806-1817.19. Morris, M.G. The effects of structure and its dynamics on the ecology and conservation of arthropods in British grasslands. Biological conservation 2000, 95, 129-142.
- Adhikari, B.; Agrawal, A. Understanding the social and ecological outcomes of PES projects: a review and ananalysis. Conservation and Society,2013,11(4),359–374.
- Hou, L.; Xia, F.; Chen, Q.; Huang, J.; He, Y.; Rose, N.; Rozelle, S. Grassland ecological compensation policy in China improves grassland quality and increases herders’ income. Nature Communications, 2021,12(1):4683. doi: 10.1038/s41467-021-24942-8.
Comment-2:
Comment-3: From my point of view, the discussion part is too long and the conclusions part to short and too specific, which must be somehow balanced. In the conclusions, in addition to summarising the actions taken and results, please strengthen the explanation of their significance. It is recommended to use quantitative reasoning comparing with appropriate benchmarks, especially those stemming from previous work, also from outside China.
Response: We have now revised the discussion and conclusion sections wherever necessary, as per suggestions.
We welcome any further suggestions from the reviewers and editors, please inform us at your kind convenience.
Best regards.
Sincerely yours,
Hui-Long Lin
E-mail: linhuilong@lzu.edu.cn
Reviewer 2 Report
The paper presents results of studies on the grassland conservation in China in relation to the Subsidy and Incentive System of Grassland Conservation (SISGC) implemented in 2011-2017. I have read the paper with interest and I think it can be published in the journal. However, I would recommend some improvements and additional explanations in the text before the final acceptance of the submission for publication. They are as follows:
- My major concern refers to the methodology, and more specifically to the sampling process. As the Authors state:
“The System Stratified Sampling (SRS), covering seven grassland types of CSCS, identified a total of 2,192 sampling points (Figure 1). Three 0.5 x 0.5m samples were randomly selected for each sampling point in the lush grassland period during 2004-2015 to capture all the aboveground green plants in the samplings into the envelope, number them and get the raw weight. (…) We got the dry weight and finally obtained the biomass data of each sampling point by calculating the average biomass data of each sampling point.” (p. 3, l. 101-111).
Firstly, it is unclear to me how many times each of these 2,192 sampling points has been visited. As I understand, there were at least two field visits: the first one before the implementation of SISGC (2004-2010), and the second one during its implementation (2011-2017), to make the samples comparable and on that basis to detect the changes in the grasslands response. Please add relevant explanations in the text.
Secondly, what do you mean by “the lush grassland period”? As it can be seen on the map (Figure 1) the sampling points are scattered over a large part of China, characterized by very different topography and climate, as you also mention in the paper. Consequently, the lengths of the vegetation periods (and the “the lush grassland periods”) also differ significantly in the respective sampling points. Additionally, I guess that you needed a considerable amount of time to get to all these sampling points, which may also affect the comparability of the respective samples (you collected the samples at different stages of vegetation periods, that is with different biomass). So, please explain: in which months did the sampling take place, and how did you ensure that the results of the biomass analyses of samples collected in (implicitly) different periods were comparable?
- The Chinese currency “Yuan” is used to describe investments related to the grasslands conservation: “The first phase involved eight major grassland regions, including Inner Mongolia, with a cumulative investment of 77.36 billion Yuan from the central government; another five provinces were added in the second phase, covering 268 pastoral and semi-pastoral counties in the country, with a planned investment of 93.8 billion Yuan from the central government.” (p. 1, l. 41-44).
I think it would be more understandable for Readers from outside China if you convert these values into some internationally recognized currencies, as for example "US dollars" or "Euros". Please reconsider and correct.
- The same is with the amount of subsidies calculated per unit area. You refer to “yuan” per “mu” of land (for example on page 18 and in Table A6), which is rather of limited use for the non-Chinese Readers. So, it is suggested to recalculate these amounts and express them in “US$” or “Euro” per “hectare” of land.
- Figures and Table A6: please change the names of the administrative units, that is “Tibet” instead of “Xizang” and “Inner Mongolia” instead of “Neimenggu”.
- Resolution of figures 2, 3, 5, 6 and 7 needs to be improved – at present is too low, making the pictures slightly unreadable.
Round 2
Reviewer 2 Report
In my opinion, explanations and improvements made by the Authors in the paper are satisfactory. This, it is recommended to accept the article for publication in present form.
